# Analysis of Road Roughness and Driver Comfort in ‘Long-Haul’ Road Transportation Using Random Forest Approach

**DOI:** 10.3390/s24186115

**Published:** 2024-09-21

**Authors:** Olusola O. Ajayi, Anish M. Kurien, Karim Djouani, Lamine Dieng

**Affiliations:** 1F’SATI, Faculty of Engineering and the Built Environment, Tshwane University of Technology, Pretoria 0001, South Africa; kurienam@tut.ac.za (A.M.K.); djouani@gmail.com (K.D.); lamine.dieng@univ-eiffel.fr (L.D.); 2LISSI Laboratory, Université Paris-Est Créteil, 94000 Créteil, France; 3MAST Laboratory, Université Gustave Eiffel, All. Des Ponts et Chaussees, 44340 Bouguenais, France

**Keywords:** road roughness, driver’s comfort, random forest, XGBoost, SVR, long haul, transportation, probe vehicle, correlation

## Abstract

Global trade depends on long-haul transportation, yet comfort for drivers on lengthy trips is sometimes neglected. Rough roads have a major negative influence on driver comfort and increase the risk of weariness, distracted driving, and accidents. Using Random Forest regression, a machine learning technique well-suited to examining big datasets and nonlinear relationships, this study examines the relationship between road roughness and driver comfort. Using the MIRANDA mobile application, data were gathered from 1,048,576 rows, including vehicle acceleration and values for the International Roughness Index (IRI). The Support Vector Regression (SVR) and XGBoost models were used for comparative analysis. Random Forest was preferred because of its ability to be deployed in real time and use less memory, even if XGBoost performed better in terms of training time and prediction accuracy. The findings showed a significant relationship between driver discomfort and road roughness, with rougher roads resulting in increased vertical acceleration and lower comfort levels (Road Roughness: SD—0.73; Driver’s Comfort: Mean—10.01, SD—0.64). This study highlights how crucial it is to provide smooth surfaces and road maintenance in order to increase road safety, lessen driver weariness, and promote long-haul driver welfare. These results offer information to transportation authorities and policymakers to help them make data-driven decisions that enhance the efficiency of transportation and road conditions.

## 1. Introduction

Since long-haul road transportation makes it possible to carry goods across great distances, it is essential to the global economy. Despite its vital relevance, the comfort of drivers during these lengthy drives is frequently disregarded. Uncomfortable driving circumstances, like those brought on by uneven roads, have been linked to driver weariness, a drop in attention span, and a higher chance of collisions, according to [1]. Due to its versatility, long-haul transportation is essential to international trade. However, long-distance driving can be taxing on drivers, compromising both their safety and well-being due to its demanding nature. Long-haul truck drivers deal with a variety of difficulties while driving for extended periods of time, including exhaustion, tension, and discomfort. Long-haul drivers’ comfort has a significant impact on their safety, degree of weariness, and general well-being [2] as discomfort can lead to fatigue, musculoskeletal disorders, and ultimately, safety risks [3].

Road roughness is a major aspect that affects the driving experience because it can affect comfort and pleasure in general on lengthy trips. This aligns with the studies conducted in [4,5], which indicated that extended exposure to unfavorable road conditions like uneven pavements may result in driver fatigue, discomfort, and even musculoskeletal diseases. The presence of potholes, bumps, and uneven surfaces on a road can greatly affect how comfortable drivers are. According to [6], the performance of drivers, job satisfaction, and general well-being are all directly impacted by their level of comfort and contentment.

Furthermore, the level of discomfort brought on by bad road conditions can raise stress levels, impair the focus of a driver, and increase the likelihood of an accident [7]. Road roughness accounts for the majority of accidents recorded as its consequential drowsy effects result into drivers losing control, driving outside the designated lane, and consequently having a head-on collision with oncoming vehicles or hitting external objects [8]. Therefore, the enhancement of road safety and maximizing the effectiveness of transportation systems require an understanding of the components, correlation and connection between road roughness and drivers’ comfort. Transportation authorities and legislators can improve road infrastructure and lessen the detrimental effects of bad road conditions on the well-being of drivers by examining how road roughness affects the comfort of drivers, as well as prioritize investments in road maintenance and contribute to infrastructure improvements as well as technological advancements. Moreover, cultivating a culture of road safety and sustainability requires advocating for and raising awareness of the significance of road conditions for driver comfort.

Understanding the connection between road roughness and driver comfort during long-distance driving is the focus of this study. The techniques used to investigate the relationships between various factors that affect a driver are frequently insufficient to fully capture their intricacies. This study’s primary goal is to investigate and clarify the intricate relationships between road roughness and the comfort of drivers during long-distance driving using the Random Forest machine-learning technique. In doing so, the study hopes to offer insights that can help with enhancing the quality of life of a driver and guiding the upkeep and improvement of road infrastructure. This study is anticipated to provide significant advances in our understanding of how road roughness affects driver comfort and increase our knowledge of the critical elements and relationships that cause discomfort when traveling long distances. The study encourages a data-driven decision-making approach by showcasing the effectiveness of Random Forest in conducting the correlation analysis, notably in capturing nonlinear correlations and handling huge, complicated datasets. In the end and not losing sight of computationally complexity issues as they relate to running time and memory consumption, this study carries out a comparative analysis of Random Forest regression, XGBoost regression and SVR regression models.

Research in the transportation sector often employs the Random Forest approach as well as other machine-learning algorithms for tasks such as route optimization, traffic prediction, and safety analysis. The benefits of using the Random Forest approach include its capacity to manage extensive and intricate datasets. Random Forest is a potent machine-learning technique that can identify nonlinear correlations among variables [9]. It handles multidimensional data and finds significant predictor variables when analyzing the relationship between road roughness and drivers’ comfort. Its capacity to produce precise forecasts and feature importance rankings makes it an excellent choice for examining the complex relationships between road roughness and driver comfort. Using Random Forest analysis, we can find subtle trends in the data that other statistical techniques would miss, giving stakeholders and policymakers in the transportation sector important new information to support decision-making. In order to provide a thorough understanding of the complex interaction between road conditions and drivers’ comfort during long-haul transportation, this study explores the use of Random Forest regression, which is adept at handling complex, nonlinear relationships [10]. Traditional methods for studying this relationship, such as surveys or simple statistical analysis, often struggle to capture the intricacies involved [11].

By leveraging Random Forest, we aim to gain a deeper understanding of how road roughness interacts with other factors to influence driver comfort in long-haul transportation. Previous studies have looked at how driving behavior, ride quality, and car performance are affected by uneven roadways. Nonetheless, this study gives voice to the issue by assessing how drivers’ comfort is affected by uneven roads. In this study, vertical acceleration data were extracted from a sensor smartphone, and the extracted data were analyzed to discuss the relationship between road roughness, driver health and road safety. Vertical acceleration data are a measurement of acceleration along the vertical axis (*z*-axis) of a vehicle. They refer to data representing the upward and downward forces acting on a vehicle (vibrations) as it reaches over road surfaces (be it bumpy surfaces, potholes, or uneven surfaces). In this study, the vertical acceleration data were captured using the MIRANDA app mounted on a BlackView smartphone.

Furthermore, the use of Random Forest regression analysis is justified by comparing its output in terms of running time and memory usage, with XGBoost and Support Vector Regression models. By utilizing cutting-edge machine-learning techniques, this work pushes toward a comprehensive knowledge of road roughness and comfort during long-haul transportation.

The study differs from the existing works in that
i.It focuses on driver comfort—an important but frequently overlooked topic in long-haul transportation research—as opposed to other studies that primarily examined vehicle performance and vibrational responses.ii.In contrast to other studies that employed linear models or more straightforward statistical techniques, this work uses a nonlinear Random Forest methodology. This allows for a more detailed knowledge of the intricate links that traditional methods could miss between comfort and driving conditions.iii.Unlike other studies that rely on smaller datasets or less-precise data sources, this one employs a large dataset collected by smartphone sensors and processed using Python.iv.A comparison analysis between Random Forest, XGBoost, and SVR is included in the study to assess the trade-offs between real-time usability, memory consumption, and model complexity—a point that has not been as heavily stressed in other publications.

Succinctly, the following are the contributions of this study:i.This study provides a data-intensive investigation into the relationship between road roughness and driver comfort by analyzing 1,048,576 rows of data obtained from probe vehicles using the MIRANDA application (3.1.0).ii.By helping to improve road infrastructure to satisfy standard road indices, the study’s findings encourage more efficient transportation networks and help standardize the methodology used for evaluating road conditions. Consequently, the study offers practical insights for policymakers to prioritize road repair, which can enhance driver well-being and road safety, by addressing how road conditions influence driver weariness.iii.The study uses IRI criteria to evaluate road roughness and driver comfort, improving our knowledge of how these elements affect long-haul driver safety and insights into the physical strain that driving conditions put on drivers.iv.The study highlights Random Forest’s compromise between memory consumption and ease of use for real-time deployment by comparing it with XGBoost and Support Vector Regression (SVR) models.

## 2. Related Work

The impact of bumpy roads on driving habits, ride quality, and automobile performance has been the subject of earlier research. To evaluate the effect of road roughness on the comfort of drivers, the related and relevant literature in this domain is discussed.

The expert-authored research study in [12] simulates the vibrational responses of a vehicle-driver system, addressing low-frequency vibrations and identifying driver model parameters through experimental data. The study employs a radial-basis neural network (RBNN) and random vibration theory to anticipate vehicle accelerations and simulate vertical vibrations. Important discoveries include the best viscous damping for shock absorbers and how well neural networks simulate complicated systems. The RBNN, with Gaussian functions, delivers quick convergence and reliable predictions. For increased accuracy, future studies should include growing RBNNs, examining other basis functions, and investigating various network topologies and learning strategies.

In order to evaluate the usefulness of the International Roughness Index (IRI) as a pavement performance indicator, the study in [13] employs a neural network to investigate the relationship between the IRI and pavement distress. It reports a good association (0.944) between distress measures and the IRI by using a back-propagation neural network to analyze distress data from pavement photos. The findings validate the usefulness of the IRI as a dependable pavement condition index and its ability to simplify road repair decisions. The paper recommends more research to improve the neural network model and assess the long-term effects of repair techniques on the IRI.

The study in [14] explores the dynamic interaction between road roughness and heavy vehicles, focusing on pavement dynamic wheel loading (DWL) and driver-perceived ride quality. Using indices like Profile Index (PIt) and ORMSPE, along with RoadRuf software and PSD analysis, the research shows that road roughness predicts DWL across various frequency bands. Low-frequency vehicle-body vibrations significantly impact ride quality. The study emphasizes considering both high- and low-frequency vibrations when predicting DWL and highlights factors like climate and soil type. It suggests further research on the effects of vegetation and maintenance history on DWL.

The authors in [15] use signal processing techniques, particularly power spectrum density (PSD) and wavelet transform (WT), to analyze road profiles from laser profilometer data. Focusing on road roughness and pavement structure, the study finds that wavelet-based analysis outperforms PSD in identifying high-frequency defects like cracks and potholes. The research shows that wavelet analysis aids in maintenance by offering insights into pavement issues and recommends further research to integrate advanced signal processing with other data sources for improved accuracy and efficiency.

In order to establish comfort limits for road users, the authors in [16] aim to establish comfort limits by linking road roughness, measured by the International Roughness Index (IRI), to the whole-body vibration (WBV) experienced by drivers. Using a vehicle model and dynamic simulations, the study found a strong correlation (R^2^ = 0.75–0.93) between weighted vertical acceleration (a_WZ) and the IRI at different speeds. The results suggest threshold values for evaluating comfort, which can guide road design and maintenance. Further research is recommended to refine models for improving road engineering.

Using an instrumented test vehicle, the authors in [17] investigate how road surface roughness affects a vehicle’s ability to identify bridge frequencies using an instrumented test vehicle. The study addresses the issue of surface roughness distorting vehicle spectra, making bridge frequency identification more difficult. Using closed-form theory and vehicle–bridge interaction simulations, data were collected from vehicle acceleration responses on bridges with varying roughness levels. The results show that rough surfaces hinder accurate bridge frequency detection, highlighting the need for countermeasures and further research to improve frequency-identification methods despite roughness.

The research work in [18] evaluates the dynamic-load increases cars impose on road surfaces due to surface irregularity vibrations, focusing on the vehicle–pavement interaction and dynamic overload caused by these irregularities. Using a quarter-car model (QCM) in Matlab^®^. It analyzes the force transferred between the pavement and different vehicle types (car, bus, truck). The results show that surface degradation can predict dynamic overload, providing insights for pavement maintenance and design. The study recommends further research on diverse road conditions and vehicle types to deepen our understanding of dynamic impacts on pavements.

The author in [19] proposes using car-mounted accelerometers and sensors to monitor pavement conditions through a new metric called the road impact factor (RIF), derived from accelerometer data. The study aims to provide a cost-effective alternative to traditional metrics like the expensive International Roughness Index (IRI). By combining power spectral density (PSD) properties with IRI localization, the RIF and time–wavelength-intensity transform (TWIT) were developed. The results showed that the RIF and the IRI are proportional, and TWIT allows for low-cost, repeatable metrics at different speeds. It suggests that the RIF and TWIT could improve pavement roughness assessment, recommending further research for practical application.

The goal of the study in [20] was to develop an efficient, low-cost system using GPS and *Z*-axis accelerometers to calculate the International Roughness Index (IRI), offering a practical alternative to traditional, costly methods. A successful small-scale test demonstrated the system’s accuracy and efficiency for pavement maintenance, with recommendations for further optimization and testing. The main objective of the authors in [21] was to develop a predictive system for evaluating pavement roughness on low-traffic highways in India, addressing the speed sensitivity of the bump integrator technique. By testing on 15 road segments at various speeds (20–50 km/h) and analyzing factors like the IRI, age, and cracking severity, the authors created conversion equations for accurate roughness readings. The study highlights the need for generalized equations to account for speed variations and recommends further research to refine the model for broader applicability.

The goal of the authors in [22] was to provide a real-time, cost-effective method for evaluating road roughness using in-pavement strain sensors. By linking sensor data to the International Roughness Index (IRI), the approach replaces traditional methods with continuous monitoring. Using GFRP-FBG sensors installed at MnROAD, the system achieved high accuracy within 3.3% of IRI estimates. The results demonstrate the reliability and affordability of in-pavement sensors for real-time road-roughness monitoring, with recommendations for further research to expand their use in transportation engineering.

Road roughness has an effect on vehicle energy dissipation and fuel consumption, as examined in the research article [23]. The study aims to measure the correlation between road roughness and vehicle fuel consumption through an analysis of rolling-resistance energy dissipation. Using a mechanistic model with random vibration techniques, the research links the International Roughness Index (IRI) and dissipated energy to road profiles and vehicle dynamics. The results show that energy dissipation scales with the square of the IRI and is influenced by road waviness. The study highlights road roughness’s environmental impact and recommends further research to refine the model and explore additional factors affecting energy dissipation. The authors in [24] examined driver comfort and dread-zone boundaries in left-turn-across-path/opposite-direction (LTAP/OD) scenarios—a major cause of traffic deaths. Using a test-track experiment with 22 drivers, data on post-encroachment time, lateral acceleration, and self-reported comfort were collected. The results showed that hurried drivers accepted larger lateral accelerations and shorter time gaps. The study introduces the concept of a “dread zone” and emphasizes the need for further research to improve road safety, particularly for driver assistance systems and autonomous vehicles.

The authors in [25] conducted a study funded by the German Federal Ministry of Transport, which focuses on evaluating three-dimensional pavement-evenness data collected by modern road-mapping systems. By applying advanced data processing techniques, the research aims to develop effective methods for assessing road-surface evenness and roughness. It analyzes statistical models to improve pavement interpretation, emphasizing the importance of accurate evaluations for enhancing driving comfort and road safety. In order to better understand how distress kinds like cracking, rutting, and raveling affect pavement roughness in the Jazan road network, the work in [26] examines the relationship between pavement condition and roughness measurements, focusing on distress factors like cracking, raveling, and rutting. Using data collection, correlation analysis, and regression modeling, the findings show that while rutting does not affect the International Roughness Index (IRI), cracking and raveling do. The study emphasizes the importance of considering different types of distress when evaluating pavement quality and ride comfort, recommending the integration of distress data and roughness measures for a more comprehensive evaluation method.

The authors in [27] introduce an affordable compact road profiler that measures road roughness in real time using GPS and accelerometers, aiming to enhance pavement management systems (PMS). By integrating data with geographic information systems (GISs), the study offers improved accuracy over traditional high-speed profilers, accounting for both inner and outer wheel paths. The findings highlight the significance of road class, local networks, and seasonal variations in PMS. The work recommends further research to refine the technology and explores its broader applications, particularly for municipal road management to improve maintenance and user satisfaction. The goal of the study in [28] is to improve the reliability and accuracy of the iDRIMS system’s International Roughness Index (IRI) estimation using frequency-domain analysis. It addresses previous issues with sensor placement and speed-related errors by applying a genetic algorithm for parameter optimization and a half-car model for vehicle modeling. Data are collected via a smartphone-based system to estimate the IRI through vehicle acceleration responses. The results show that the process is robust and accurate across various road conditions. The study suggests adopting this method for practical road-condition evaluations and exploring its scalability to different vehicles and road profiles.

The study in [29] investigates the impact of different pavement types (DP, DPSA, DPSGTA) and pavement age on accident risk on urban expressways, measured by accidents per vehicle kilometers traveled (VKT). Using data on accidents, road maintenance, traffic, and weather, the study reveals that pavement age significantly affects accident risk, with friction, surface roughness, and permeability playing key roles. The findings aim to enhance road safety and inform more effective pavement management strategies.

Using the Roadroid smartphone app and a conventional inertial profiler, the study paper in [30] evaluates polyurethane-stabilized concrete pavement using the Roadroid smartphone app and a conventional inertial profiler to measure roughness. Both tools were compared using ProVAL software to generate the International Roughness Index (IRI). The results show that while Roadroid offers a cost-effective option for less-critical assessments, it underestimates roughness due to fluctuations, with the standard profiler providing more accurate results. The study concludes that Roadroid is useful for lower-cost, objective evaluations, but traditional profilers are necessary for high-traffic roads. Further research is recommended to improve smartphone app accuracy for pavement evaluation.

In order to assess road surface conditions, the study in [31] uses response-based methods. Its goals are to provide a summary of contemporary approaches, highlight variations, and suggest future research emphasis areas. To address the lack of a thorough analysis of response-based techniques currently used for road-pavement applications, the study collected papers from the previous 15 years utilizing databases such as Scopus and Google Scholar. It looked at methods for measuring road profiles and detecting anomalies using onboard sensors and Internet of Things (IoT) technology. It discovered that machine-learning methods have potential but are limited by their reliance on data. The analysis of 130 publications showed that crowdsourcing, data aggregation, and GPS accuracy are enhancing algorithms for roughness index calculation and pothole identification. The results point to the necessity of more studies in the areas of fleet vehicle algorithms, active suspension systems, and raising estimation accuracy. In order to improve road surface assessment, the study’s conclusion makes recommendations for further research on GPS accuracy, data fusion, and crowdsourcing issues.

Effective road maintenance is crucial for social, economic, and environmental benefits. The International Roughness Index (IRI) is commonly used to measure road roughness due to its link to road-use costs. With traditional methods being costly, smartphone accelerometers and GPS data are being explored as affordable alternatives for real-time road assessments. Recent advancements in smartphone applications, influenced by vehicle type, speed, and other factors, show promise. Machine learning and multivariate regression models have demonstrated potential for accurate road-roughness estimation, aiding in strategic and cost-effective road management [32]. The goal of the study in [33] is to develop a machine-learning pipeline using in-car sensors—vertical acceleration and driving speed—to estimate road roughness. It addresses the need for ongoing road monitoring due to deterioration. Various supervised machine-learning models, including support vector machines and neural networks, were tested and shown to accurately forecast road roughness. The study concludes that these models are promising for future pavement condition monitoring and recommends further research on real-time implementation, dataset expansion, and advanced approaches for improved road maintenance and passenger safety.

The work in [34] examines the relationship between road roughness, measured by the International Roughness Index (IRI), and whole-body vibration (WBV) at different vehicle speeds and road types. Data collected from nine cars over 1860 km showed that overall vibration values were significantly lower when using RMS of frequency-weighted acceleration compared to vertical seat vibrations alone. The findings stress the importance of using real vehicle vibration data and considering vehicle speed and road classification for accurate IRI thresholds and improved passenger comfort. The study recommends further research on various road conditions and vehicle types. The study in [35] examines the durability and quality of asphalt pavement, emphasizing its role in supporting foot and vehicle activity. It explores the use of inexpensive smartphone sensors to collect roughness data on various road networks, highlighting the need for smooth surfaces to improve ride quality and reduce deformation. Data collected from 15 locations using Huawei Nova 2i phones and the Roadbounce technique showed that state roads had a poorer ride quality compared to expressways. The study finds Roadbounce to be efficient and cost-effective for roughness measurements and recommends comparing smartphone-based roughness data with industry-standard profilers for accurate repair scheduling and assessment.

The research in [36] aims to forecast mean free-flow speed (FFS) on urban arterials using machine-learning algorithms, including Random Forest (RF), Support Vector Machine (SVM), and Artificial Neural Networks (ANNs). By analyzing data from 11 urban arterials and testing on two roads, the study demonstrates that ML models outperform traditional linear regression in predicting FFS and understanding traffic dynamics. The results highlight the potential for improved road planning, traffic control, and safety, recommending adoption by local governments and transportation agencies. Future research should expand data sources to enhance model generalizability across different pavement conditions. In order to improve ride comfort, the authors in [37] explore improving ride comfort by analyzing vehicle vibration characteristics and using the ISA algorithm to develop speed plans that mitigate discomfort from uneven roads. Machine-learning techniques were used to create a road recognition system and comfortable speed algorithms, with simulations showing enhanced comfort on A-grade roads and less on B- and E-grade roads. The ISA-based speed strategies significantly improved ride comfort, highlighting the impact of road slopes on passenger comfort. The study recommends further research to optimize these strategies for various road conditions and vehicle types.

The study in [38] aims to develop a mathematical model that correlates vibrations from engines and pavement roughness to better predict the International Roughness Index (IRI). By examining vehicle responses to pavement irregularities, the study uses a quarter-car model and analytical techniques like power spectral density analysis and Laplace transforms. The new model, considering variables such as driving speed and sample rate, demonstrates improved accuracy over existing methods. The study enhances our understanding of vehicle–pavement interactions and suggests further research to test the model in different scenarios. In [39], Hanandeh’s study shows that pavement management systems using artificial neural networks (ANNs) and genetic algorithms (GAs) effectively forecast and evaluate pavement conditions. The Pavement Quality Index (PQI) integrates metrics like the Ride Quality Index (RQI) and Pavement Condition Index (PCI). Enhanced ANN models with additional hidden layers and GA techniques improve accuracy in predicting the International Roughness Index (IRI) and optimizing maintenance. The combination of ANNs and GAs offers superior forecasts and decision-making compared to traditional methods, as evidenced by case studies in Mecca and Medina.

The authors in [40] propose a cost-effective method for monitoring road pavements using vibration data and video from an e-bike and a sedan. They use smartphone apps at varying speeds and iterations, highlighting the impact of speed and frequency on data accuracy. The study suggests this approach for improving road maintenance and recommends further research on factors like weather, traffic, and equipment precision. The work in [41] presents an unsupervised learning system for road condition monitoring using public transit buses as mobile sensors. GPS and accelerometer data, combined with k-means and SOM algorithms, detected damaged road areas and assessed roughness. Tested on 1150 km of bus routes in Gujarat, India, they offer a scalable, cost-effective solution for road maintenance and repair scheduling.

In order to support sustainable transportation planning, the research in [42] predicts the International Roughness Index (IRI) for asphalt pavement on arterial roads in Sri Lanka using machine-learning models, including Random Forest and XGBoost. Pavement age and traffic volume were key predictors, with SHAP analysis enhancing model transparency. Random Forest outperformed conventional methods, demonstrating AI’s superiority in forecasting road roughness, aiding timely repairs, and promoting sustainability. The study recommends further research to improve prediction accuracy with advanced methods. In [43], Ali et al. develop models using Multiple Linear Regression (MLR) and Artificial Neural Networks (ANNs) to estimate the International Roughness Index (IRI) based on pavement distress in various climate regions. Using data from the Long-Term Pavement Performance (LTPP) database, the study shows that both models accurately predict the IRI, with ANNs outperforming MLR. The findings highlight key relationships between the IRI and pavement distress, suggesting that ANN models can enhance pavement management and maintenance decisions. Further research is recommended to refine these models.

With a special emphasis on two-wheelers, the study in [44] analyzes the effects of pavement and geometric design factors on run-off road crashes on rural two-lane curves. Using crash data, pavement conditions, and geometric features, and applying Generalized Estimating Equations (GEEs), it finds that cross-slope, sight distance, and curve radius significantly affect crash frequencies. The study recommends incorporating these design factors into road safety interventions and suggests further research to explore other variables influencing crash rates. The problem of subjective and manual evaluations in gravel-road maintenance is addressed in the study by [45]. The study developed a method for evaluating gravel-road conditions by combining audio and visual data from GoPro cameras and recordings in Sweden. The OR gate decision-level fusion method was found to be most effective. This approach allows for objective, real-time road assessments using affordable devices like smartphones, offering a precise and cost-effective solution for gravel-road maintenance with potential for community involvement and future innovations.

The literature demonstrates a restricted use of particular machine-learning methods, such as the Random Forest approach, to examine the connection between road irregularity and driver comfort. Although a number of different algorithms are discussed, not enough is known about how Random Forest in particular might offer insights into passengers’ comfort levels when traveling long distances. Thus, by examining vertical acceleration data, this study focuses on the ability of Random Forest algorithms to forecast and evaluate driver comfort. Driving on rough roads requires more attention and constant adjustments and results in physical strain, leading to quicker fatigue and exhaustion, especially during long drives. The literature also shows that no analytical attention has been paid to this perspective.

## 3. Methodology

To ensure an effective evaluation that produces the desired result, this study adopts subjective and objective assessment approaches. For the objective assessment, we employ MIRANDA, a mobile application developed at the University of Gustave Eiffel, France. The application, which is loaded on a smartphone, is mounted on a driver’s vehicle and captures and records the acceleration of the vehicles’ movements as well as other variables. For the subjective assessment, the study subjected the data collected through objective assessment to further computation using Python scripts to calculate the magnitude of the acceleration from which the road roughness level was determined. The driver’s comfort level was also computed from the acceleration values (x, y, and z) alongside the acceleration magnitude. The data captured from these two assessments were then subjected to Random Forest machine-learning analysis to evaluate the road roughness level and predict the driver’s comfort level. Using many decision trees and combining their predictions, Random Forest, as a supervised machine-learning technique, increases accuracy and robustness. In order to model the association between road roughness and drivers’ comfort, the analysis was further subjected to Pearson and Spearman correlation analysis to evaluate the correlation between the driver’s comfort and the status of the road.

Table A1 and Table A2 (Appendix A and Appendix B) show the data collected and the model computed from them, while Figure 1 is the system model.

### 3.1. Detailed Analysis of the Model’s Components

#### 3.1.1. Initializing and Loading MIRANDA App

The device used for measurement is an Android smartphone. The device utilized was a 6.5-inch Blackview BV9200 Rugged Smartphone (Paris, France) running Android 12 with 256 GB of RAM, a 50 Megapixel camera, and 3G/4G connectivity. Raw data were provided by the sensors included in most smartphones (time, acceleration, GPS locations, etc.). The MIRANDA (Measurement of Road Indicators by Nomadic devices) application was installed on the smartphones. The measurement session was managed by the MIRANDA application, which also handled the settings, survey activation and deactivation, measurement file production, and other tasks. During the test drive, the smartphone was integrated into a probe vehicle and utilized to gather data. Over the course of the data collection, eight (8) different probe vehicles were used. From these, 5,242,880 rows of data were generated; however, due to the maximum number of rows an Excel file could return, only 1,048,576 rows of data could be captured and used for this study.

#### 3.1.2. The Server and Database

The gathered data were sent to a back-end server, where the data were automatically analyzed to produce an estimated road profile and the associated indicator. An uploaded database contained the completed data/information. The finished data/information were uploaded to a database.

#### 3.1.3. Converted and Merged Zipped Data

The data generated through the MIRANDA app and the sensor-based smartphone were stored as zip files https://filesender.renater.fr/?s=download&token=64a52d74-9a45-4747-9c47-9859095abcb1 (accessed on 23 July 2024). Figure 2 shows the codes used to extract the zip files, while Figure 3 show the scripts written to convert and also merge the various zip files/folders.

#### 3.1.4. Preprocessing and Feature Selection

The collected data were further processed using data cleaning techniques (removal of incomplete/missing data). A filter feature selection technique was deployed to remove noisy or unwanted/irrelevant features. Figure 4 and Figure 5 show the Python scripts used in the preprocessing and direct selection of variables/columns from the study’s dataset.

#### 3.1.5. Data Splitting

Data splitting involves partitioning the dataset into subsets to enhance training and testing. For this study, the dataset was divided into a training dataset and testing dataset in the proportion of 80:20. Figure 6 is the Python code snippet used for the splitting.

#### 3.1.6. Model Initialization, Training, Prediction, Evaluation and Computations of Running Time and Memory Usage

As a common practice in machine learning, the random initialization seed value is 42, to ensure the consistency and reproducibility of results. The model training (X_train and y_train) was specified to allow the model to learn how the independent variables relate to the dependent variables. To avoid overfitting, the prediction was carried out on x_test data. This also assisted in ensuring that our model generalizes to new, unseen data. Performance evaluation (using Mean Squared Error (MSE), Mean Absolute Error (MAE) and R-Squared (R^2^) metrics) was also carried out by predicting based on the x_test data.

The key complexity challenges in RFR, which are memory usage and training time, were managed by tuning key parameters, optimizing the Random Forest with limited depth and a random selection of data points. Memory profilers, using psutil library and *time module*, were deployed in the codes for optimizing the memory usage and running time, respectively. Figure 7 shows the Python codes executed to carry this out and also some other outputs.

#### 3.1.7. Correlation Analysis and Outputs

Figure 8 and Figure 9 show the Python scripts for the correlation analysis and plots, respectively, using both Pearson and Spearman’s correlation types.

## 4. Results and Discussion

### 4.1. The Model

As stated in Section 3.1.7, the Random Forest regression model was evaluated using the following metrics: Mean Squared Error (*MSE*), Mean Absolute Error (*MAE*) and R-Squared (*R*^2^) (Equations (1)–(3)).
(1)$$MSE=\frac{1}{n}\sum_{i=1}^{n}\left(y_{i}-\hat{y_{i}}\right)2$$
where
*n*—number of data points/observations (1,048,576 rows of data).$y_{i}$—actual value of the target variable (accl_magnitude) for the *i*-th data point.${\overbrace{y}}_{i}$—predicted value of the target variable (accl_magnitude) for the *i*-th data point.$\left(y_{i}-\hat{y_{i}}\right)2$—squared difference (or error) between actual and predicted values for each data point.

(2)$$MAE=\frac{1}{n}\sum_{i=1}^{n}\left|y_{i}-\hat{y_{i}}\right|$$
where
*n*—number of data points/observations (1,048,576 rows of data).$y_{i}$—actual value of the target variable (accl_magnitude) for the *i*-th data point.$\overbrace{y_{i}}$—predicted value of the target variable (accl_magnitude) for the *i*-th data point.$\left|y_{i}-\hat{y_{i}}\right|$—absolute difference (or error) between actual and predicted values for each data point.

(3)$$R^{2}=1-\frac{SSres}{SStot}$$
where
SS_res_-$\sum_{i=1}^{n}\left(y_{i}-\hat{y_{i}}\right)2$—the residual sum of squares.SS_tot_-$\sum_{i=1}^{n}\left(y_{i}-\hat{y_{i}}\right)2$—the total sum of squares.*n*—number of data points/observations (1,048,576 rows of data).$y_{i}$—actual value of the target variable (accl_magnitude) for the *i*-th data point.$\overbrace{y_{i}}$—predicted value of the target variable (accl_magnitude) for the *i*-th data point.$\bar{y}$—mean value of the actual values of the target variable.

The results computed show

Evaluating Random Forest...
Random Forest Regressor Training Time: 135.4263 secs
Random Forest Regressor Mean Squared Error: 0.0006
Random Forest Regressor Mean Absolute Error: 0.0157
Random Forest Regressor R Squared: 0.9958
Random Forest Regressor Peak Memory Usage: 337.7812 MB


Evaluating XGBoost...
XGBoost Regressor Training Time: 6.1867 secs
XGBoost Regressor Mean Squared Error: 0.0002
XGBoost Regressor Mean Absolute Error: 0.0060
XGBoost Regressor R Squared: 0.9987
XGBoost Regressor Peak Memory Usage: 65.3438 MB

Evaluating SVR...
Support Vector Regressor Training Time: 10.0060 secs
Support Vector Regressor Mean Squared Error: 0.0006
Support Vector Regressor Mean Absolute Error: 0.0187
Support Vector Regressor R Squared: 0.9954
Support Vector Regressor Peak Memory Usage: 381.2227 MB

Total Program Memory Usage:
Total Memory Used (RSS): 338.1602 MB

As could be seen from the results, the XGBoost regression model outperformed the Random Forest regression (RFR) model, being the model with the lowest errors (MSE: 0.0002; MAE: 0.0060), the highest R-squared (0.9987) and the fastest training time (6.15 s), while RFR turns out to be the most memory-effective model.

Vividly, XGBoost might have outperformed RFR, and the use of RFR as the major model for the analysis of this study can be further justified as follows:i.While XGBoost can be faster during training, RFR has quicker prediction times in deployment, for real-time or near-real-time solutions.ii.RFR handles noisy data and outliers better, due to its ensemble nature.iii.For the nature and peculiarity of this research and its practicality, RFR’s simplicity and lower computational cost during deployment are key advantages.

### 4.2. The Metrics

Road roughness is commonly measured using the International Roughness Index (IRI), which is a commonly recognized standard. Road roughness is measured in meters per kilometer (m/km), with rougher roads denoted by greater numbers. Table 1 and Table 2 show the comparison between the IRI values and relevant acceleration measures, such as the acceleration magnitude (accl_magnitude) and standard deviation of vertical acceleration (accl_z), for road roughness and driver comfort levels, as presented in this study.

The study’s computed results:


Road Roughness Level (Standard Deviation of accl_z): 0.7288071585301953
Driver’s Comfort (Mean of accl_magnitude): 10.0101394667174
Driver’s Comfort (Standard Deviation of accl_magnitude): 0.6411496881655732

The Road Roughness Level (Standard Deviation of accl_z) of 0.7288 falls within the Moderately Rough category, which corresponds to an IRI value between 2 and 4 m/km. This suggests that the road condition is average, with noticeable vertical movement.

The Driver’s Comfort (Mean of accl_magnitude) of 10.0101 m/s^2^ and (Standard Deviation of accl_magnitude) of 0.6411 m/s^2^ places the ride in the Uncomfortable category. This corresponds to an IRI value between 4 and 6 m/km. This indicates that the road conditions are rough enough to cause discomfort during driving.

### 4.3. The Correlation Matrix

From Figure 9a, we note that there is a moderate negative correlation (−0.48) between accl_x and accl_y, meaning that increases in accl_x typically lead to decreases in accl_y. The association between accl_x and accl_z is weak (−0.14), and there is a modest inclination for accl_z to decrease as accl_x grows. The very poor positive correlation (0.10) between accl_x and accl_magnitude suggests that there is hardly any linear relationship between the two variables. For accl_y and accl_z, a moderately positive correlation (0.32) indicates that there may be some relationship between increases in accl_y and rises in accl_z. Increases in accl_z are strongly correlated (0.74) with increases in accl_magnitude, suggesting a strong positive relationship between the two variables. For time and other variables, there is no significant linear relationship, as evidenced by the extremely weak correlations (from −0.02 to 0.26) with the acceleration components.

In a nutshell, the matrix shows that there is a substantial correlation (0.74) between the variables accl_z and accl_magnitude, suggesting a close relationship between them. Furthermore, accl_x and accl_y have a moderately negative association (−0.48), while accl_y and accl_z have a moderately positive correlation (0.32). On the other hand, time has weak correlations with the other variables, indicating that it does not have a significant and linear impact on the acceleration readings.

### 4.4. Implication of Findings

The results of this study have important ramifications for transportation agencies and decision-makers in government. By prioritizing smoother road surfaces and minimizing discomfort and safety hazards for long-haul drivers, infrastructure investments and maintenance strategies can be guided by an understanding of the influence that road roughness has on drivers’ comfort. Additionally, the application of Random Forest analysis highlights the usefulness of machine learning in managing complicated datasets and capturing nonlinear correlations, both of which can be extended to other transportation-related research fields. This study emphasizes the potential for improving driver well-being, lowering accident rates, and fostering a culture of road safety and sustainability through better road conditions.

## 5. Conclusions, Recommendation and Suggestions for Further Research

By clarifying the complex relationship between road roughness and drivers’ comfort, this study adds to the expanding corpus of research on road transportation. The study’s conclusion highlights how important road conditions are in affecting long-haul drivers’ comfort and safety. The use of Random Forest regression, in comparison with XGBoost and SVR models, shows that while XGBoost offers slightly better predictive power and performance, the trade-off in simplicity and robustness gives RFR an edge as a more viable method in practice. It is, however, advised that future works should employ an ensemble method to achieve better performance.

As part of the recommendation of this study, transportation authorities should give road maintenance and infrastructure upgrades first priority in order to promote driver well-being and lower safety hazards, especially in areas that largely rely on long-haul transportation. To obtain a more thorough understanding of the variables affecting driver comfort, it is suggested that future studies should investigate the inclusion of further factors, such vehicle type, speed, and driver characteristics, in their analysis.

Furthermore, the broadening of this study to include other road conditions and geographic areas can offer a more comprehensive understanding of the findings’ worldwide application. The usefulness of cutting-edge technologies, including real-time road monitoring systems, in reducing the negative effects of uneven roads on driver comfort may also be the subject of future study.

## Figures and Tables

**Figure 1 sensors-24-06115-f001:**
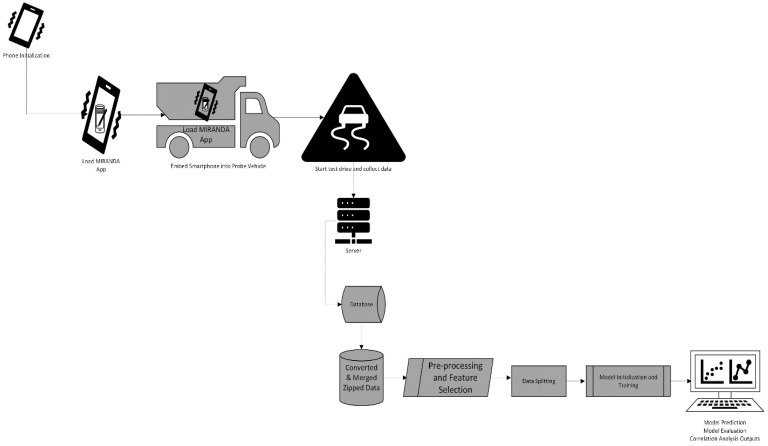
System model.

**Figure 2 sensors-24-06115-f002:**
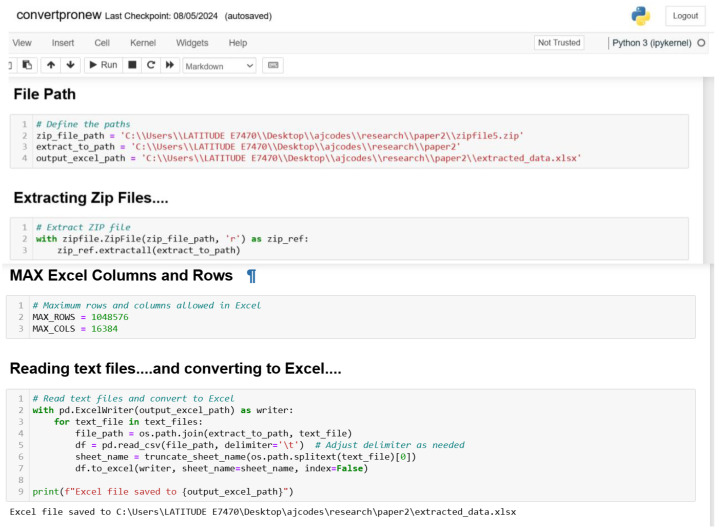
Python file extraction scripts.

**Figure 3 sensors-24-06115-f003:**
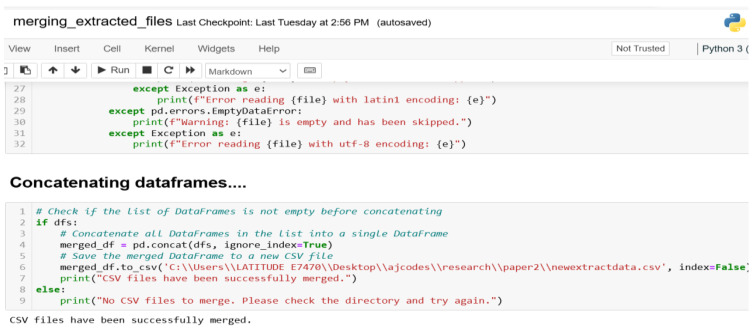
Python file merge scripts.

**Figure 4 sensors-24-06115-f004:**
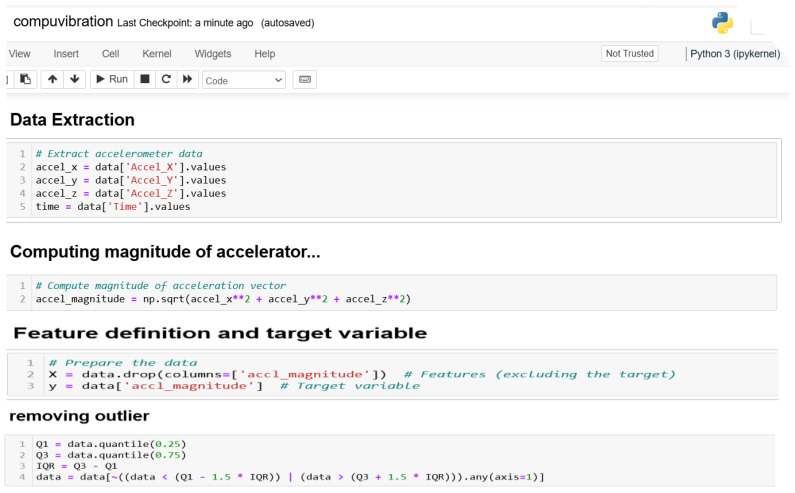
Data preprocessing.

**Figure 5 sensors-24-06115-f005:**
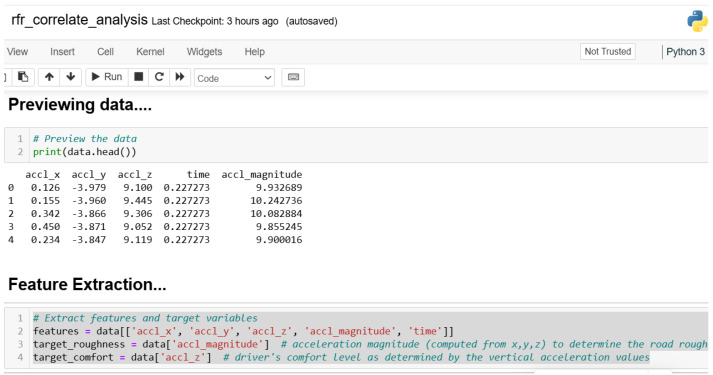
Feature selection.

**Figure 6 sensors-24-06115-f006:**
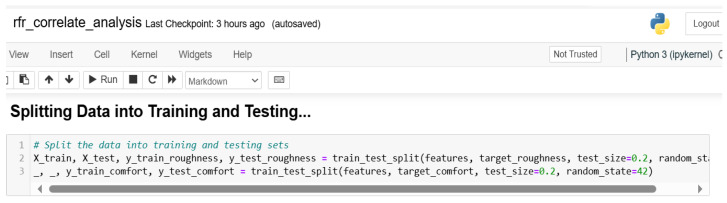
Data splitting.

**Figure 7 sensors-24-06115-f007:**
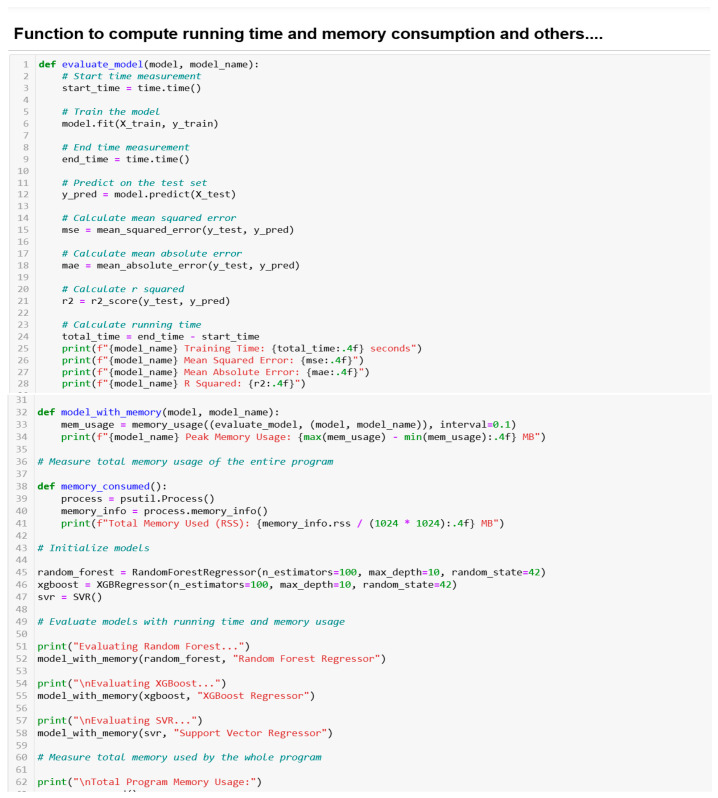
Model initialization, training, prediction, evaluation and computations.

**Figure 8 sensors-24-06115-f008:**
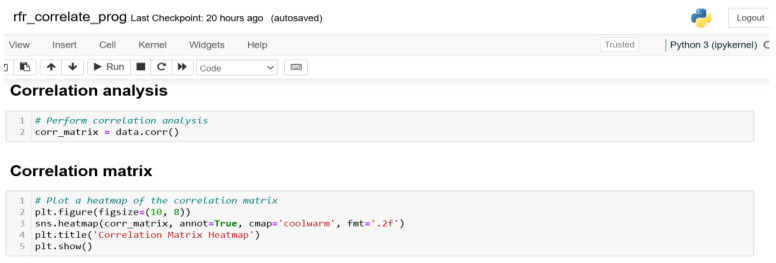
Correlation analysis.

**Figure 9 sensors-24-06115-f009:**
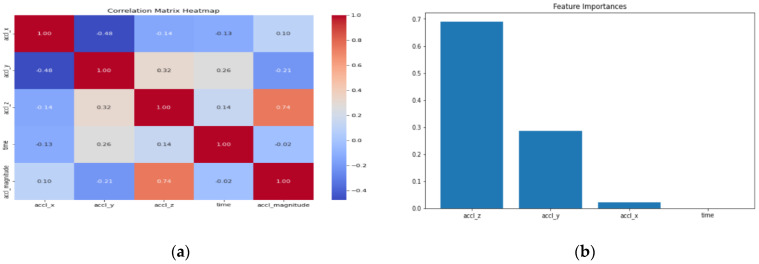
(**a**): Correlation matrix. (**b**): Bar chart.

**Table 1 sensors-24-06115-t001:** The IRI vs. road roughness level.

IRI Classification	IRI Value (m/km)	Standard Deviation of Vertical Acceleration (accl_z)	Road Condition Description
Very Smooth	0–1	<0.3	Excellent road condition, minimal vertical movement.
Smooth	1–2	0.3–0.5	Good road condition, slight vertical movement.
Moderately Rough	2–4	0.5–0.8	Average road condition, noticeable vertical movement.
Rough	4–6	0.8–1.2	Poor road condition, significant vertical movement.
Very Rough	>6	>1.2	Very poor road condition, severe vertical movement.

**Table 2 sensors-24-06115-t002:** The IRI vs. driver’s comfort level.

Comfort Level	IRI Value (m/km)	Mean Acceleration Magnitude (accl_magnitude) (m/s^2^)	Standard Deviation of Acceleration Magnitude (accl_magnitude) (m/s^2^)	Comfort Description
Very Comfortable	0–1	<0.2	<0.1	Extremely smooth, minimal vibrations.
Comfortable	1–2	0.2–0.5	0.1–0.3	Generally smooth, low vibrations.
Acceptable	2–4	0.5–1.0	0.3–0.6	Slightly rough, moderate vibrations.
Uncomfortable	4–6	1.0–1.5	0.6–1.0	Rough, noticeable vibrations.
Very Uncomfortable	>6	>1.5	>1.0	Very rough, strong vibrations.

## Data Availability

Data are contained within the article.

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
