# Peer review of "Analysis of Road Roughness and Driver Comfort in ‘Long-Haul’ Road Transportation Using Random Forest Approach"

_sensors, 2024, doi:10.3390/s24186115_

Round 1

Reviewer 1 Report

Comments and Suggestions for Authors

The importance of the problem addressed in this study is high. It is easy to follow the text.
 However, a few questions are not well addressed in the study:

1. In the related work there is a sentence “Thus, by examining vertical acceleration data, this study focuses on the ability of Random Forest algorithms to forecast and evaluate driver comfort.” It is the first time the vertical acceleration data is mentioned in the study. It should be clearly explained what it stands for and where the acceleration data comes from. Probably, it should be done in the "Introduction" section rather than in the short paragraph closing the Related work.

2. The authors should clearly define the contribution of the current work, preferably in the form of a list, and present it at the end of the “Introduction” section. Each statement/contribution should be clearly explained – what important issue it resolves, how is it different from existing research, why are you focusing on the addressed issue, and what you suggest.

3. The authors should present a numeric evaluation of their proposed random forest model with the existing relevant works. The experiments can be significantly improved.

4. There is no information about the developed model’s complexity, including running time, and required memory.

  Comments on the Quality of English Language

Good level of English.

Author Response

Please, see the attachment. Thank you.

Reviewer 2 Report

Comments and Suggestions for Authors

The data collected using an objective assessment was further processed using Python scripts to calculate the acceleration value, from which the level of roughness of the road was determined. The driver's comfort level was also calculated from the acceleration values along with the acceleration value. The data obtained from these two assessments were then analyzed using the random forest method to assess the level of road roughness and predict the driver's comfort level. By using multiple decision trees and combining their predictions, random forest as a controlled machine learning method improves accuracy and reliability. To model the relationship between road roughness and driver comfort, the analysis was additionally subjected to Pearson and Spearman correlation analysis to assess the correlation between driver comfort and road condition.

The results obtained show a strong correlation between road roughness and driving comfort, which can be used as an analysis of the condition of the road surface.

However, it should be noted that the literature review in the presented work is excessively extensive and detailed. It is recommended that the authors significantly shorten this section, focusing on the most significant publications directly related to the research topic. In addition, there is no analysis of the limitations and disadvantages of the method used.

In general, I believe that this work has a scientific novelty, is of interest to the scientific community and meets the requirements of the journal.

Author Response

Please, see the attachment. Thank you.

Round 2

Reviewer 1 Report

Comments and Suggestions for Authors

Previous comments resolved. No further suggestions.